# Tsunamites versus tempestites: Various types of redeposited stromatoporoid beds in the Devonian of the Holy Cross Mountains (Poland), a case study from the Ołowianka Quarry

**Piotr Łuczyński** [ID]*

Faculty of Geology, University of Warsaw, Warsaw, Poland

* Piotr.Luczynski@uw.edu.pl

## Abstract

The sedimentary history of two stromatoporoid accumulations – an allobiostrome and a parabiostrome–are studied in the shallow water carbonates of the Middle to Upper Devonian Kowala Formation in the Ołowianka Quarry, Holy Cross Mountains, central Poland. Sedimentological and facies observations are accompanied by morphometrical and taphonomical analyses of redeposited stromatoporoid skeletons. Stromatoporoid features, including shape profile, latilaminae arrangement, surface character, dimensions, and preservation state, are interpreted in terms of their original growth habitats and susceptibility to exhumation and transport. Sedimentary features of the studied beds are interpreted with regards to the high-energy processes that lead to their deposition. In the allobiostrome, the original stromatoporoid habitat was located below storm wave base, in a calm setting characterised by a low and stable depositional rate and clear bottom waters. The large scale onshore redeposition of stromatoporoid skeletons from such a setting was only possible due to an extraordinary event causing erosion at considerable depths: a tsunami is the most probable explanation. The sedimentary and textural features of the allobiostromal accumulation, such as clast supported textures and lack of vertical sorting, point to a single act of deposition and high flow velocities, in agreement with the tsunami interpretation. In contrast, the parabiostromal stromatoporoid accumulation does not exhibit any features that would require a non-tempestitic explanation, the default and most probable interpretation of high energy facies interbedding shallow water lagoonal sediments. This comparison has shown that studies of variously developed stromatoporoid beds, and particularly the analysis of morphometric features of stromatoporoid skeletons, can provide a unique opportunity to identify palaeotsunamites, which commonly remain undetected in the sedimentary record, leading to underestimates of their abundance.

**Citation:** Łuczyński P (2022) Tsunamites versus tempestites: Various types of redeposited stromatoporoid beds in the Devonian of the Holy Cross Mountains (Poland), a case study from the Ołowianka Quarry. PLoS ONE 17(5): e0268349. https://doi.org/10.1371/journal.pone.0268349

**Data Availability Statement:** The minimal dataset is within the paper. The studied thin plates illustrated on Fig 7 are deposited and available in the S.J. Thugutt Geological Museum, Faculty of

Geology, University of Warsaw, Poland (MWGUW)
under the inventory number: MWGUW 009779.

**Funding:** The study was financed from the funds of
the Faculty of Geology, University of Warsaw. (geo.
uw.edu.pl) Grant number 501-0113-01-1130203.
The funders had no role in study design, data
collection and analysis, decision to publish, or
preparation of the manuscript.

**Competing interests:** The authors have declared
that no competing interests exist.

## Introduction

In the Middle and Late Devonian, large areas of the southern shelf of the Laurussia continent (Euramerica) were occupied by vast carbonate platforms [1, 2]. The best exposures of Eifelian to Frasnian shallow-water carbonate successions are known from the Ardennes [3, 4] and the Rheinish Slate Mountains [5, 6]. To the east, deposits of the Devonian shelf fringing Laurussia are best exposed in the Holy Cross Mountains of central Poland. The Devonian succession of the Holy Cross Mountains generally follows the same pattern as that of the Ardennes [7], with carbonate facies replacing clastic sedimentation in the Eifelian (middle Devonian) and a deepening pulse drowning the shallow-water carbonate platforms around the Frasnian-Famennian boundary [8–10]. However, in contrast to the Ardennes, which are characterised by the prominent development of reefs and mounds in the Frasnian [11–13], large carbonate buildups in the Holy Cross Devonian are relatively rare [14–18]. Instead, the Givetian and Frasnian platform successions of the Holy Cross Mountains abound in a variety of "stromatoporoid-coral limestones", a large proportion of which are developed as various types of stromatoporoid biostromes [9, 19–21]. In the present paper, a peculiar example of such beds in the abandoned Ołowianka Quarry is studied and described.

There are a wide variety of stromatoporoid biostromes, *sensu* Kershaw [22], in the Holy Cross Devonian, ranging from autobiostromes with *in situ* stromatoporoids, through parabiostromes, composed mostly of redeposited and reworked skeletal material, to allobiostromes, built entirely of allochthonous and partly destroyed skeletons [23, 24]. The sedimentological features of these stromatoporoid biostromal beds hold considerable information on the processes governing their deposition in shallow-water platform settings, especially in conjunction with the ecological and taphonomical characteristics of the redeposited stromatoporoid material. Evidence of large scale redeposition indicates the occurrence of high-energy sedimentary events that influenced shallow water sedimentation. The factors controlling stromatoporoid biostrome growth have been studied to the greatest extent in the Silurian biostromes of Gotland [25–27]; in the Devonian of the Ardennes, they were described, for instance, by Da Silva and others [28, 29]. Stromatoporoid beds of the Holy Cross Devonian were recently analysed in detail, most notably by the author of this paper, who concentrated on stromatoporoid morphometrical features and interpreted them in terms of their controlling environmental factors [18, 21, 23, 24].

Stromatoporoid reefs and biostromal accumulations that are, in many aspects, similar to those of the Holy Cross Devonian, are a typical component of the Upper Silurian succession of the Podolia region in western Ukraine [30–32]. They represent a zone of shoals and barriers along a carbonate shelf that fringed the Baltica continent (the East European Craton) with exposures ranging from Ukraine through the Baltic states to Gotland [27, 33, 34]. In the Podolian succession, parabiostromal accumulations interbedded within shallow-water lagoonal sediments have been interpreted as resulting from high-energy sedimentary events driving onshore redeposition of skeletal material [30, 35]. Some of these biostromes composed of broken and redeposited specimens were described as tsunami deposits [36].

The possibility of a palaeotsunami record in the shallow-water Upper Devonian deposits of the Holy Cross Mountains has been previously suggested by Kaźmierczak and Goldring [37], who based this premise on their interpretation of flat pebble conglomerates deposited in a subtidal setting. This is especially intriguing given that flat pebble conglomerates of an alleged tsunami genesis also accompany the stromatoporoid parabiostromes of Podolia [36]. That being said, the high energy facies (mainly developed as variously coarse-grained deposits ranging from calcarenites to carbonate breccias, commonly with abundant bioclastic material) that are interbedded upon, and fringe the flanks of, the Devonian shallow-water carbonate platform deposits of the Holy Cross Mountains, have usually been interpreted as associated with storms

[38–40] or gravitational phenomena [41, 42], rather than with tsunamis. While these interpretations are well-founded, and I do not intend to devalue nor negate them, the present paper attempts to present a possible tsunami interpretation of an allobiostromal stromatoporoid accumulation exposed in the Ołowianka Quarry. Studies of morphometrical and taphonomical attributes of stromatoporoid skeletons, combined with sedimentological analysis of biostromal beds, enable the identification of diagnostic differences between a typical parabiostrome of probable storm origin and an allobiostrome interpreted as a palaeotsunamite.

## Geological setting

The Ołowianka Quarry is located in the southwestern portion of the Holy Cross Mountains in central Poland (Fig 1A). Structurally, the exposure belongs to the northern limb of the Chęciny anticline [43, 44] (Fig 1B), and represents the easternmost extent of Devonian (and, more broadly, Palaeozoic) exposures in the Holy Cross Mountains. It lies between the village and hill of Miedzianka to the southwest, and the large, active Ostrówka Quarry to the northwest (coordinates: 50.8483 N, 20.3712 E). The Devonian limestones of Miedzianka are famous for their rich copper and lead mineralization [45, 46], while the Ostrówka Quarry is one of the largest in the Holy Cross region, exposing the whole Upper Devonian and lowest Carboniferous succession [10].

The Palaeozoic of the Holy Cross Mountains consists of two tectonic units, the southern Kielce unit and the northern Łysogóry unit, separated by the Holy Cross Dislocation (Fig 1A). The two regions differ distinctly with regards to both the thickness and development of particular stratigraphic units and their stratigraphic completeness. In terms of facies development, the Devonian of the Holy Cross Mountains is divided into three main zones: the northern Łysogóry zone, the central Kielce zone, and the southern Chęciny-Zbrza zone [47]. The central Kielce zone generally represents the shallowest facies in the area, consisting of a Givetian-Frasnian bank to reef complex flanked from the north and south by relatively deeper intrashelf basins [9].

The Ołowianka Quarry lies in the Kielce zone in both the tectonic and facies sense – that is, the southern Kielce tectonic unit and the central Kielce palaeofacies zone, respectively. In the latter context, it is located in the central Kielce (Dyminy) region (Fig 1C). In terms of lithostratigraphy, the succession exposed in the quarry belongs to the Kowala Stromatoporoid-Coral Dolomite and Limestone Formation, which, in the central Kielce region, attains a maximum thickness of up to 800 m [20]. It is divided into several beds and members that subtly differ in their development; all, however, represent a shallow-water carbonate platform environment. According to Racki [9], the succession exposed in the Ołowianka Quarry corresponds to an interval ranging from the Early Givetian to the Middle Frasnian (Fig 2).

In the present paper, the lower part of the succession, which most probably represents the Stringocephalus Beds (Middle Givetian; Fig 2), is examined in detail. In the central (Dyminy) part of the Kielce palaeofacies zone, the Stringocephalus Beds are not separated from the overlying Lower Sitkówka Beds by the Jaźwica Member, as is the case in exposures located in more southern palaeofacies. As such, due to the absence of the Jaźwica Member, the lack of reliable conodont stratigraphy in these very shallow water facies, and the general resemblance of the Stringocephalus and Sitkówka Beds, it is difficult to demarcate the exact lithostratigraphic boundary between these beds in the quarry, and thus exclude the possibility that the upper part of the described section lies within the Lower Sitkówka Beds.

## Morphometrical and taphonomical analysis of stromatoporoids, and utility in palaeoenvironmental reconstructions

Stromatoporoids, together with tabulate and rugose corals, are among the most common biotic components and framework builders of Palaeozoic organic bioherms, and biostromes

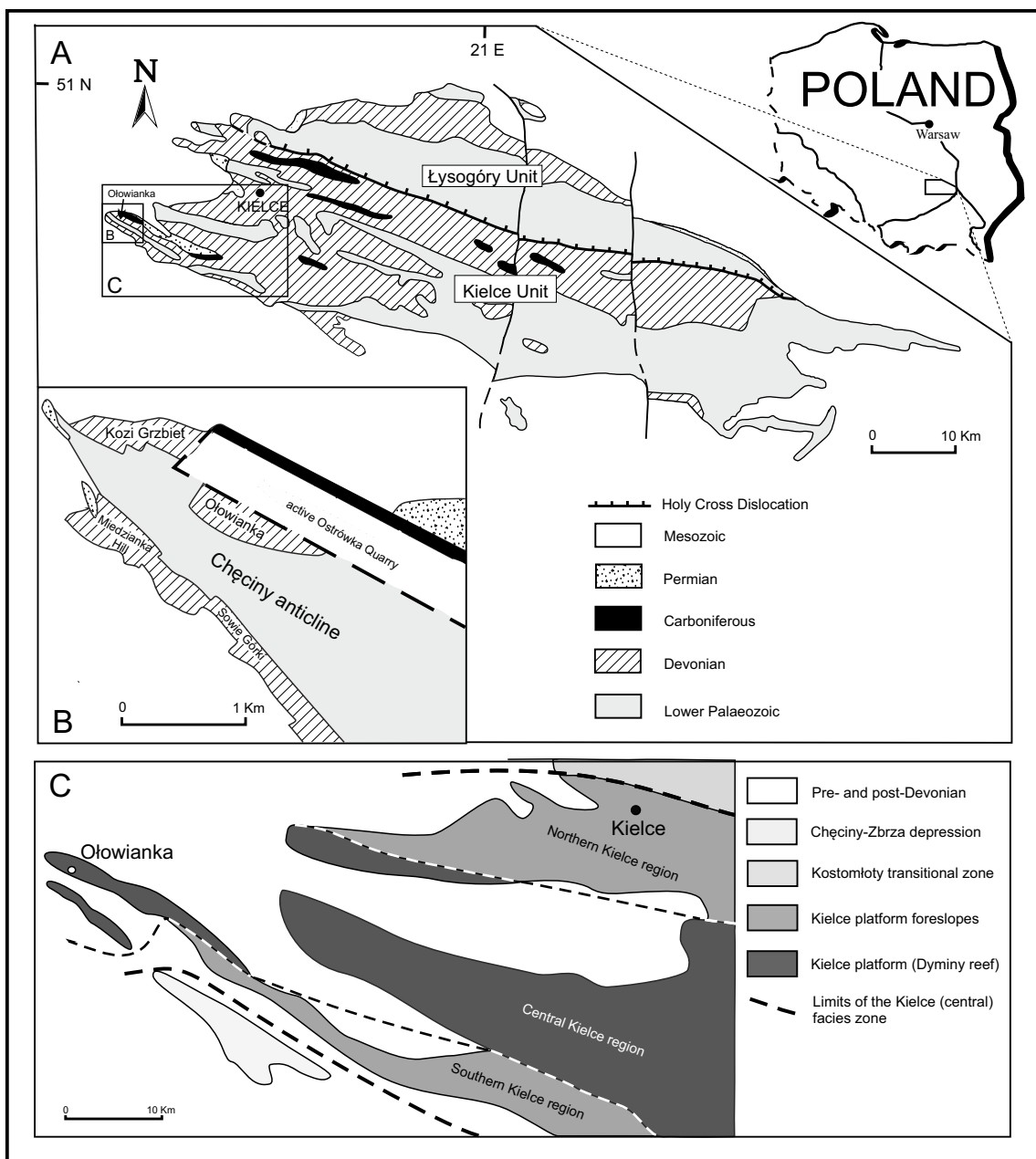

**Fig 1. Location of Ołowianka Quarry.** A – position on a sketch geological map of the Palaeozoic of the Holy Cross Mountains; B – position on a detailed geological map of the western part of the Chęciny anticline, in the vicinity of Miedzianka hill (modified after [44] under a CC BY license, with permission from [Ann Soc Geol Pol], original copyright [1975]); C – Givetian to Frasnian palaeogeography of the western part of the Kielce unit (modified after [9] under a CC BY license, with permission from [Acta Palaeontol Pol], original copyright [1993]).

[48–51]. Stromatoporoids grew in a wide array of shallow-water habitats, including reefs and bioherms, but were also commonly constituents of level-bottom communities on carbonate banks and shelves, the latter mostly preserved in the stratigraphic record as biostromes [52–54]. If not bound microbially, in most cases stromatoporoid buildups did not possess rigid frameworks, but rather were composed of densely packed, but nonetheless loose, specimens [55]. Their soft-substrate platform dwelling ecological strategy [56, 57] enabled the

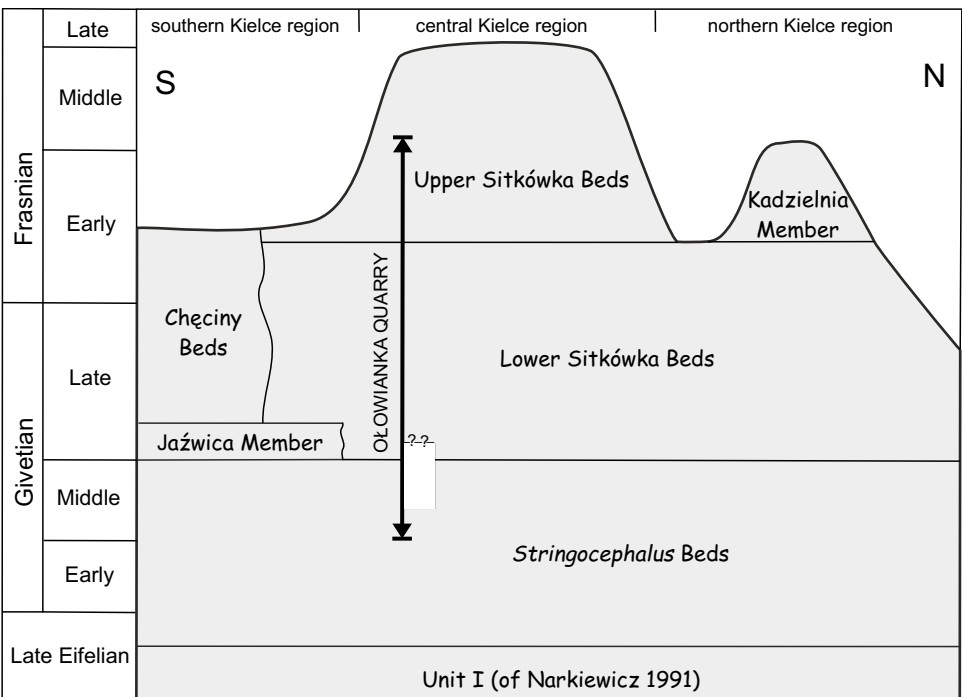

**Fig 2. Stratigraphic position of the studied Ołowianka Quarry succession within the Kowala Formation.** White rectangle represents the succession illustrated on Fig 5. (modified after [20] under a CC BY license, with permission from [Geol Q], original copyright [1990]).

colonization of vast shallow-water areas, however, rendered them highly susceptible to exhumation and redeposition. Consequently, in combination with their relatively low specific weight, which enhanced stromatoporoid buoyancy [58, 59], many Silurian and Devonian parabiostromal accumulations built of redeposited material are composed almost solely of stromatoporoids [27, 30, 36, 54, 60].

The massive basal skeletons of Palaeozoic stromatoporoids are known to adopt a wide range of forms. Their external shapes and other attributes are, in several aspects, primarily governed by environmental factors [18, 57, 61], and thus stromatoporoids are often considered to be useful palaeoenvironmental indicators [62–65]. Particular macroscopic morphometrical features of massive stromatoporoids have been previously interpreted, for instance, with respect to depositional rate and dynamics, water turbulence, substrate consistency, and seafloor inclination [for a detailed reappraisal of the factors controlling stromatoporoid morphology, see 61, 66].

In order to systematically describe the observed variety of stromatoporoid shapes, Kershaw and Riding [67] introduced a stromatoporoid morphology parameterization, which has been widely adopted, improved, and supplemented [18, 24, 52, 61, 68, 69]. The shape of massive basal stromatoporoid skeletons ranges from *laminar*, through *domical* (divided into low-, high-, extended-, and highly extended domical), to *bulbous* (divided into low- and high bulbous) (Fig 3A). In practice, the classification of a specimen as a particular shape is based on the proportions of three dimensions measured in horizontal, vertical, and diagonal aspects on a vertical crosscut through a complete skeleton. Other than the shape itself, the most important macroscopic morphometric features interpretable in terms of environmental conditions are: upper surface character, which can be *smooth* or *ragged*; the arrangement of latilaminae (major growth bands), which can be *enveloping* and *non-enveloping* (with intermediate

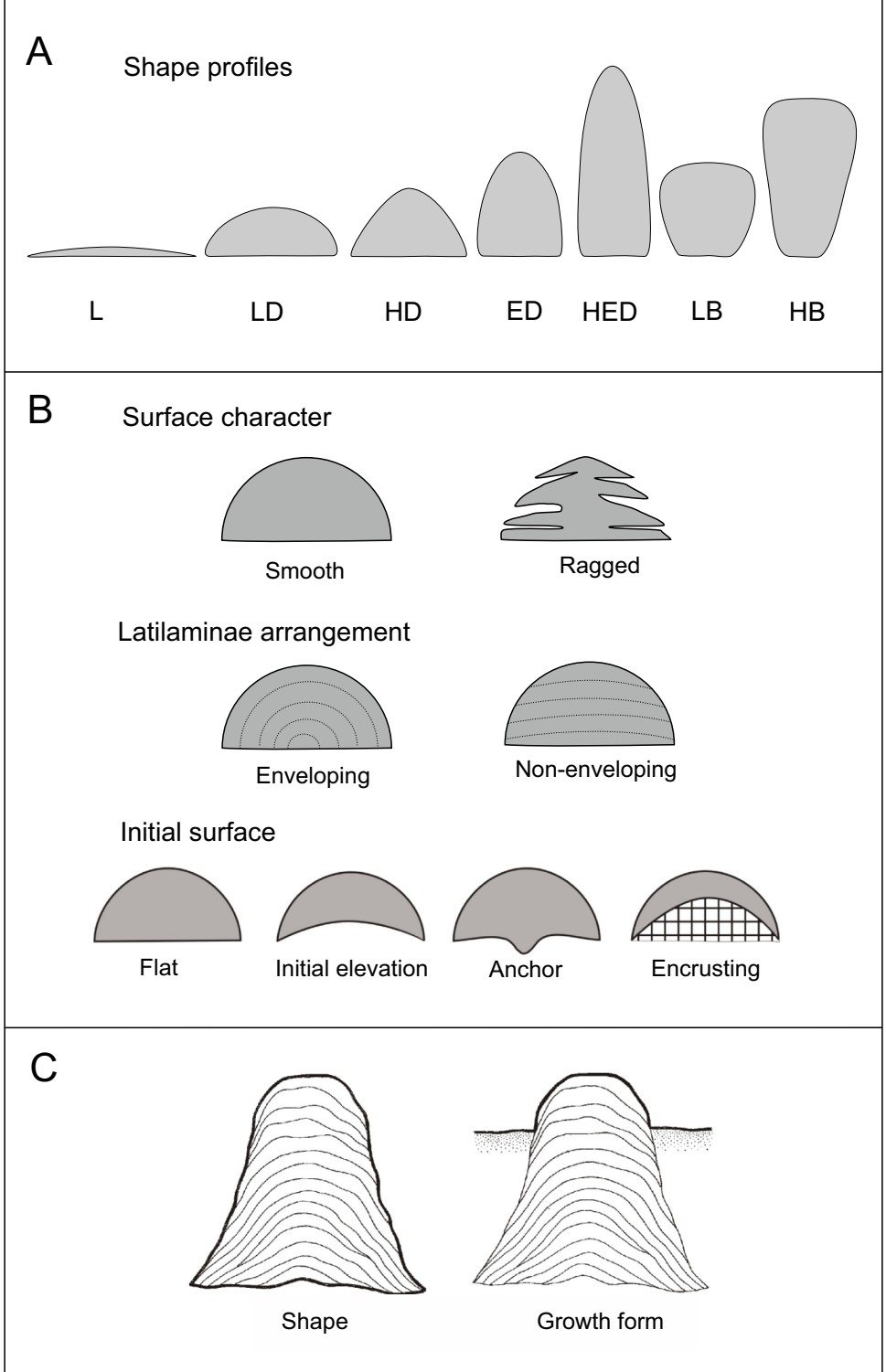

**Fig 3. Basic stromatoporoid shapes and morphometric features (after [61]).** A–Basic morphotypes: laminar (L), low domical (LD), high domical (HD), extended domical (ED), highly extended domical (HED), low bulbous (LB) and high bulbous (HB); B – Basic macroscopic morphometric features; C – Distinction between the stromatoporoid shape and the growth form (living surface profile).

variants); and the type of initial surface, which is referred to as *flat*, *initial elevation*, *anchor*, and *encrusting* (Fig 3B). An analysis of latilaminae arrangement enables workers to discern skeletal shape from the living surface profile (i.e., growth form), which represents the part of the specimen protruding over the sediment surface (Fig 3C); the relation between the two is described by the burial ratio (*BR*) parameter [69]. More sophisticated measurements, designed to quantify the shape of the upper surface (for instance, upper surface curvature and convexity) and the shape and inclination of the skeletal sides, have been recently introduced [61]. These protocols, however, require excellently preserved material and thus, as in the current study, commonly cannot be employed.

Considering the susceptibility of stromatoporoids to redeposition, and that, in many cases, stromatoporoid skeletons are preserved in allochthonous accumulations, a combined approach to the study of their morphometrical features is required. Information on both the original growth environment of particular stromatoporoids and the factors that caused their transport and final deposition in parabiostromal accumulations can be revealed by combined ecological and taphonomical analyses, in conjunction with sedimentological and facies studies.

## Reconstruction of stromatoporoid growth environment based on morphometrical and taphonomical features

As is the case for all sessile benthic organisms, stromatoporoid skeletons grew in direct interaction with the environment and the surrounding accumulation of sediments. Their latilaminar structure (i.e., the existence of repetitive growth bands) is commonly considered to reflect annual banding [66, 70–72]. The growth rates of stromatoporoids are estimated to typically lie between 1 and 3 mm/year [73, 74], which suggests typical skeletons represent several dozen or hundred years of growth (in the case of particularly big specimens [17], potentially even thousands of years), as calculated based on the number of latilaminae. During their protracted life-spans, individual stromatoporoids recorded changing environmental and sedimentary conditions via their growth patterns.

It is usually difficult, and potentially even pointless, to pinpoint an individual, pivotal environmental factor responsible for a given form of stromatoporoid skeleton [61]. Nonetheless, some typical suites of features can, with a reasonable degree of certainty, be ascribed to specific sedimentary conditions.

With regards to environmental factors, the rate and style of deposition have the most direct impact on specimen shape [21, 23, 24, 52, 54, 57, 61, 66, 72, 75, 76]. Stromatoporoids grew as sediments accumulated around them; thus, usually only parts of their upper surfaces remained unburied and were covered by soft living tissue from where growth could continue. Therefore, the final skeletal shape and the changing growth form protruding from the seafloor in successive growth stages (Fig 3C) are two distinct features with different implications for paleoenvironmental analyses [69]. Enveloping latilaminae arrangements (with subsequent latilaminae completely covering those preceding and protruding to the bottom of specimens; Fig 3B), in conjunction with smooth upper surfaces, indicate low deposition rates and, usually, calm sedimentary settings. Such stromatoporoids, which remained uncovered by sediments during their growth, are referred to as erect [18]. A non-enveloping latilaminae arrangement is characteristic for semi-buried stromatoporoids [18]. The style of sediment accumulation around such stromatoporoids can be inferred from the character of their upper surfaces and flanks–punctuated, in the case of ragged varieties, and continuous in the case of smooth. In some cases, margin raggedness has also been interpreted to reflect small overhangs protruding a few millimetres above the sea floor, with small cavities beneath [77]. Depositional rate and character, and their relation to the tempo of stromatoporoid growth, can also be inferred directly

from the burial ratio, which quantifies the proportion of the skeleton protruding above the sediment surface. The hazard of potential burial *via* sedimentation, and thus the necessity to keep pace with rapid deposition, was probably the most important factor influencing stromatoporoid growth. Initial elevations, high profiles, and ragged margins (and in particular, the co-occurrence of these characters) are the clearest indications of such conditions, in addition to internal sediment increments interrupting skeletal growth [52, 65, 69, 71].

Local variations in depositional rates can often be related to water column turbidity, directional water flow, and local seafloor palaeotopography. These factors also influenced stromatoporoid growth and can be inferred from stromatoporoid shapes [57, 63, 74, 78], especially in the case of asymmetrical specimens [18, 24, 75].

Another growth environment parameter that influenced the external shape of stromatoporoids is substrate consistency [19, 66, 76, 77]. A variety of features, such as low profiles (especially in the early phases of skeletal growth, as revealed by latilaminae arrangements), initial elevations on which growth began, and side protrusions forming overhangs with cryptic fauna beneath, have been interpreted as adaptations to muddy substrates. Furthermore, growth in muddy bottom waters also introduced the hazard that stromatoporoid pores could be clogged by tiny, suspended sediment particles [61]. Upper surface convexity, low burial ratios of final living surface profiles, and smooth upper surfaces combined with non-enveloping latilaminae arrangements are the most obvious responses to such conditions.

## Reconstruction of biostrome sedimentary environment based on stromatoporoid morphometrical and taphonomical features

Stromatoporoid morphometric features can be interpreted not only in terms of environmental and growth conditions during life, but can also provide insight on the processes of skeletal exhumation, transport, deposition, and burial in para- and allobiostromal accumulations [30, 35, 36].

Various stromatoporoid forms demonstrated different susceptibility to redeposition [53]: some morphotypes, such as the bulbous form, were easily torn out of sediments, while others, such as the low domical form, required higher current velocities to be disturbed. The type of initial surface may also have been of some importance: specimens with an anchor or initial elevation were more effectively bound to the substrate than those with a flat lower surface [23], especially in the early stages of skeletal growth. In the case of larger specimens, the most important character describing stromatoporoid vulnerability to exhumation is the burial ratio, which reflects the proportion of the skeleton protruding from the sediment during growth [69]. Additionally, the total size and weight of the skeleton had a considerable influence [18].

A slightly different suite of features describes stromatoporoid susceptibility to long-distance transport. After skeletal exhumation, features describing stromatoporoid relation to the sediment, such as the burial ratio or type of initial surface, did not have a major impact on ease of transportation. Instead, the specimen's weight and dimensions becomes the most important factor; differences in stromatoporoid sizes in various secondary accumulations can be interpreted with regards to gravitational selection during transport. Additionally, overall shape had a considerable influence on transport susceptibility: equidimensional forms, such as bulbous or high domical, were more readily transported in traction than flat or elongated forms, such as laminar. The final depositional conditions of para- and allobiostromal stromatoporoid accumulations may, additionally, be inferred based on their sedimentary attributes. Stromatoporoid bed thickness, skeletal packing, size and shape sorting, and graded bedding all can provide clues as to the ultimate depositional environment. Taphonomical characteristics, such as breakage along latilaminae or at the edges, are additional features that can be analysed with regards to transport and depositional conditions.

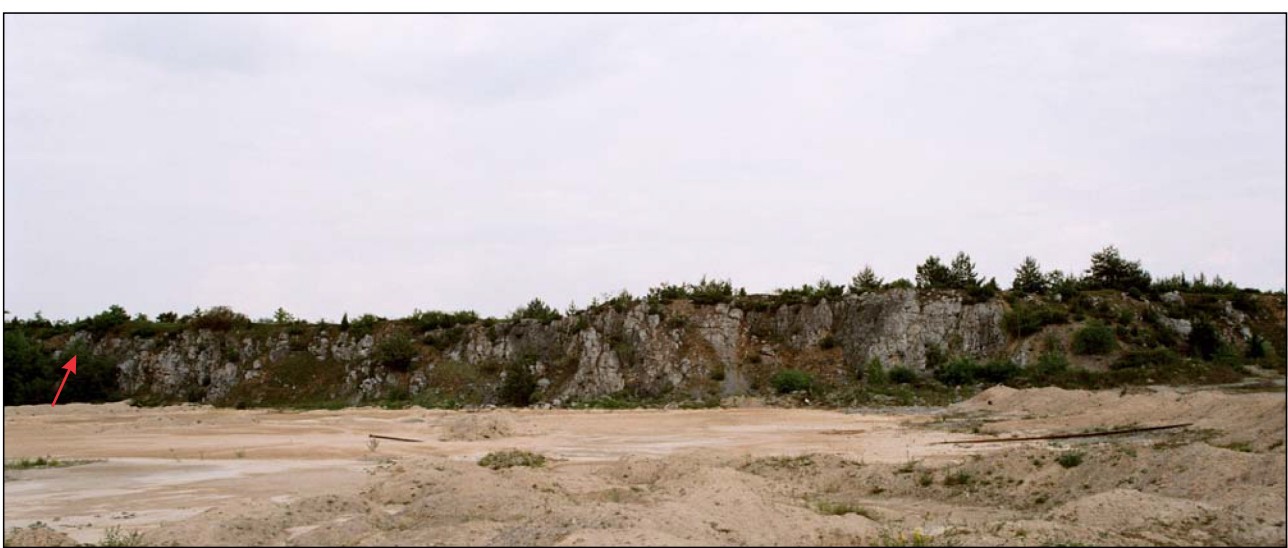

**Fig 4. Ołowianka Quarry–general view of the northern wall.** The arrow corresponds to the position of the allobiostrome.

## Materials and methods

Fieldwork was conducted in the abandoned Ołowianka Quarry. Exploitation in the quarry ceased in the 1980s: today, it is state owned, barren land with free access to the general public. According to Polish law, no permits are required to perform geological studies in such an area. The present study concentrated on the lower 80 metres of the succession exposed in the quarry (Fig 4), through which a continuous section can be prepared (Fig 5). The higher part of the succession is only partially accessible to observation.

The described Stringocephalus Beds section in the Ołowianka Quarry was examined *via* standard facies and microfacies studies. Special attention was paid to the occurrence and taphonomy of stromatoporoids with massive basal skeletons in various shapes, and to accompanying branching forms such as *Amphipora* and *Stachyodes*.

All the basic macroscopic morphometric attributes of massive stromatoporoid colonies were determined. However, as most specimens are visible on exposed weathered surfaces, some of which are not completely accessible, and dissected in random crosscut orientations, in most cases it was impossible to properly conduct measurements following the parameterization method. To avoid excluding the vast majority of specimens from consideration, stromatoporoid shapes were determined only by ascribing the specimens to the most general categories–laminar (visibly flat), domical (with basal length larger than the height, subdivided into low- and high domical, with basal length more than twice and less than twice the height, respectively) and bulbous (with height larger than the basal dimension). On the other hand, other stromatoporoid macroscopic features important for paleoenvironmental reconstructions, such as latilaminae arrangements and characters of the upper and lower surfaces, were readily apparent in the field due to outlining *via* the subtle weathering of exposed rock surfaces in an abandoned, but recently active, quarry. This also enabled the identification of stromatoporoid position and orientation, through which *in situ* and redeposited specimens could be distinguished, as well as the broken and incomplete nature of some skeletons.

In some cases, particular attributes could not be determined, and were labelled as undetectable or undetermined. Only specimens considered to be complete (or almost complete) were taken into consideration for palaeoecological analysis.

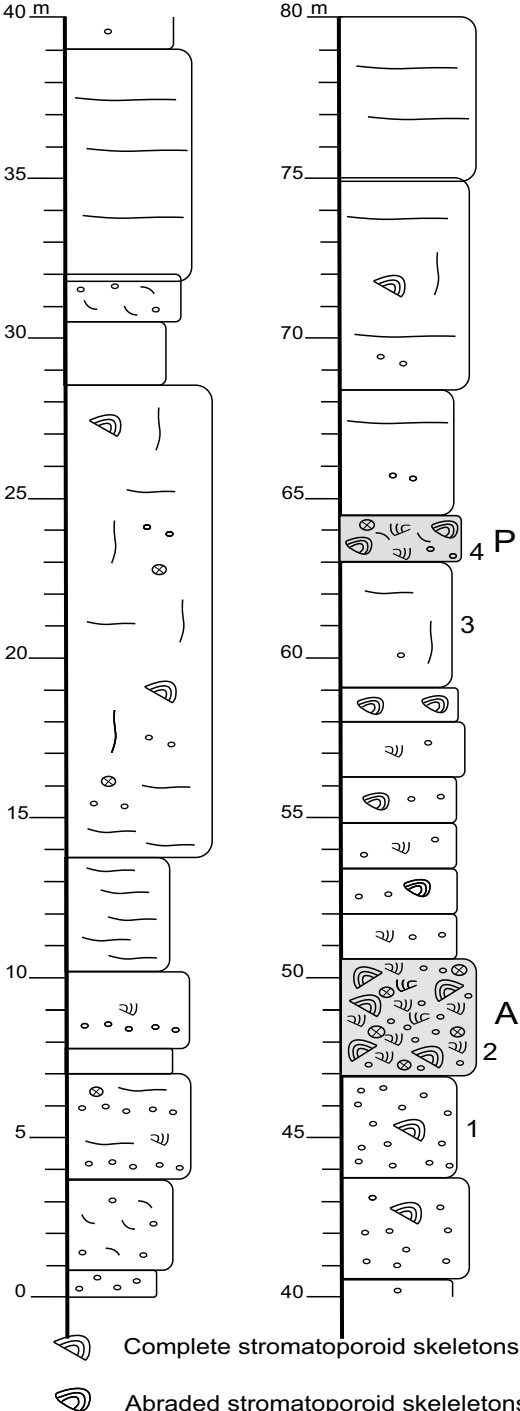

**Fig 5. Lithological section exposed in the Ołowianka Quarry.** A–allobiostrome, P – parabiostrome; 1–4 –microfacies samples illustrated in Fig 7.

The main focus of the present study is on the thick biostromal stromatoporoid accumulation exposed in the upper part of the described section (A on Fig 5), referred to as "the allobiostrome" (Fig 6A–6D). Its sedimentary features and the morphometrical and taphonomical attributes of constituent stromatoporoids were compared with a thinner stromatoporoid parabiostrome, exposed over a dozen meters higher in the profile (P on Fig 5), referred to as "the parabiostrome" (Fig 6E–6F). Stromatoporoids are also scattered through a number of other beds, mainly in upper part of the exposed section (Fig 5): however, beyond the two named beds, they do not form biostromal accumulations (and thus are referred to as non-biostromal beds). The basic morphometric features of 379 stromatoporoid skeletons were analysed here; 273 from the main allobiostrome, 69 from the main parabiostrome, and 37 from other beds. Both specimens found *in situ* and those that were overturned and redeposited, if the basic morphometric features could be determined, were taken under consideration.

Stromatoporoids from the Ołowianka Quarry were studied by Kaźmierczak [19], and more recently by Wolniewicz [79]. The latter ascribed the specimens from Ołowianka to the Late Givetian *Actinostroma-Clathrocoilona* association, which also occurs in the nearby Sowie Górki and Stokówka localities. Within this association, *Actinostroma* is by far the most abundant genus among the massive forms, and is accompanied by *Hermatostroma*, *Stromatopora*, and other genera in smaller proportions. However, there is no information as to what kinds of accumulation – allobiostromal, parabiostromal, or other–individual specimens represent (or from which locality they were sampled). Based on their shapes – for the most part, laminar and low domical [79] – they most probably do not represent the main allobiostrome, but rather were sampled from non-biostromal accumulations (see discussion). That being acknowledged, the associations from both the allobiostrome and the parabiostrome generally reflect the same set of stromatoporoids, with a distinct dominance of *Actinostroma*.

All sedimentary features of stromatoporoid accumulations, as well as the results of morphometrical and taphonomical observations, were analysed and interpreted in terms of possible high-energy sedimentary events that lead to their deposition.

## Results

### Facies and microfacies of the Stringocephalus Beds

The succession exposed in the Ołowianka Quarry is generally developed as grey, faintly bedded to massive limestones (Fig 5). The dominant pelitic limestones are, in places, interlayered by marly beds. Most beds contain scattered *Amphipora* (abundant in some horizons, forming meadows), and at times are accompanied by rare massive stromatoporoids and corals. The exposed succession is typical of the Stringocephalus Beds, which represents the bank stage of Devonian facies development in the Holy Cross area [9], and corresponds to generally calm shallow water settings with weak open marine influences. Against this background, the stromatoporoid biostromal beds represent episodes of substantially higher energy.

The microfacies of the Stringocephalus Beds are mainly represented by micritic/microspar peloidal mudstones (Fig 7C and 7D), and amphiporoid peloidal wackestones to packstones with numerous *Amphipora* and *Stachyodes* (Fig 7A and 7B). The matrix of the stromatoporoid allobiostrome developed as stromatoporoid-detrital rudstones with a densely packed, grain-supported texture composed of stromatoporoid fragments (Fig 7E and 7F), whereas by contrast the matrix of the parabiostrome consists of stromatoporoid detrital limestones (grainstones and rudstones) with stromatoporoid fragments, *Amphipora*, and shell debris (Fig 7G and 7H).

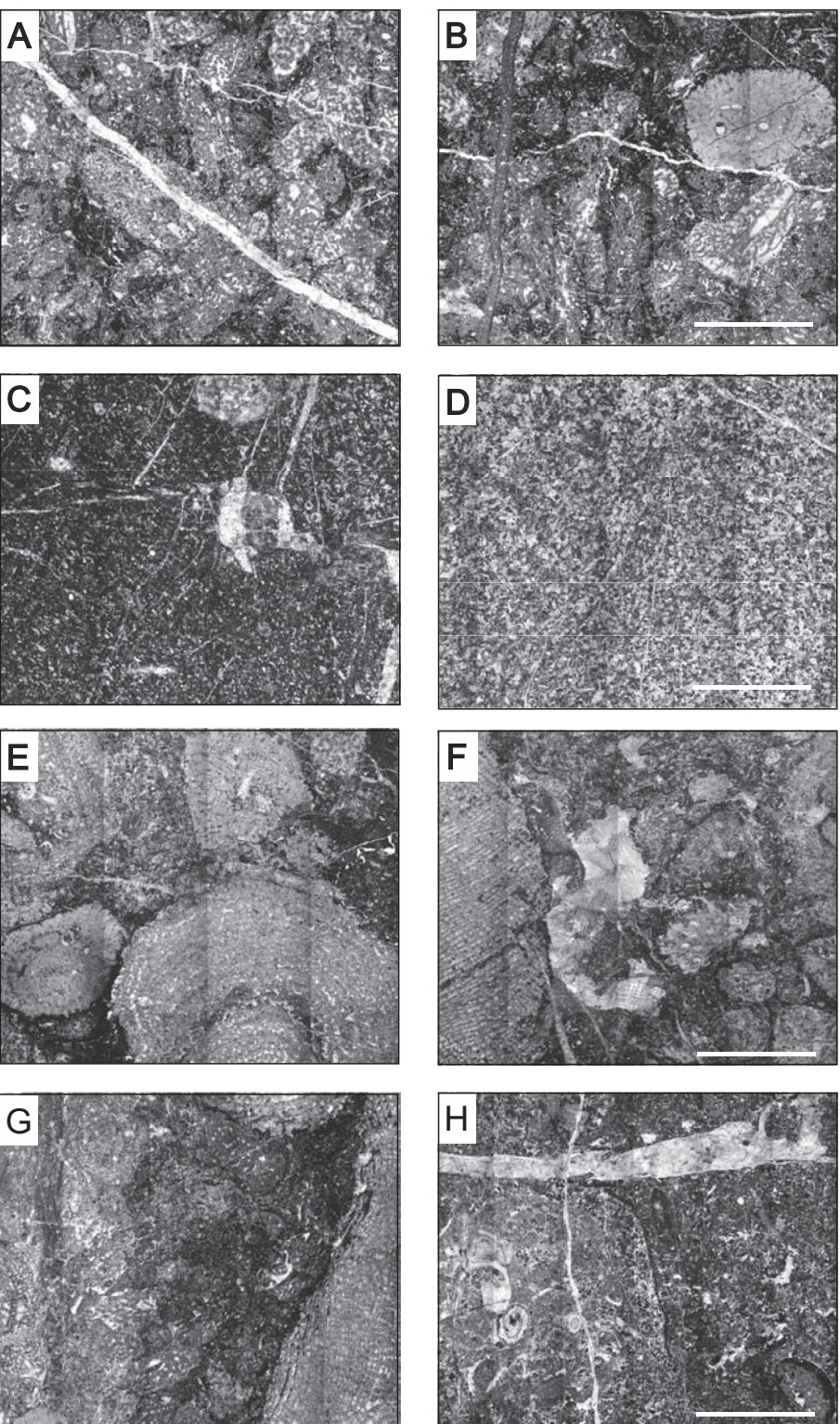

**Fig 6. Stromatoporoid bearing beds.** A-D–allobiostrome; E, F–parabiostrome. The wall in A is approximately 4 m high, the visible part of the wall in D is approximately 3 m high.

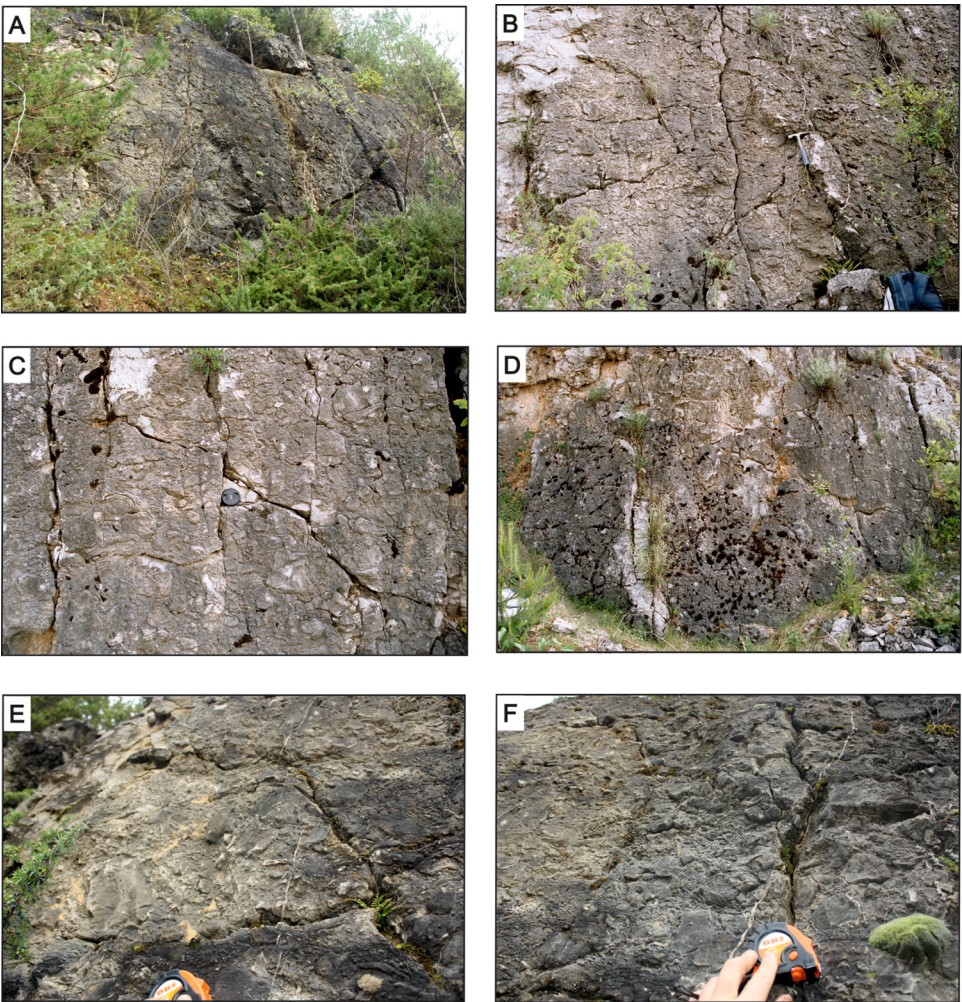

**Fig 7. Microfacies of the Stringocephalus Beds and the stromatoporoid-bearing allo- and parabiostromal accumulations.** A, B–amphiporoid peloidal limestones (wackestones-packstones) with numerous *Amphipora* and *Stachyodes* (sample 1 on Fig 5); C, D–micritic/microspar peloidal limestones (mudstones) (sample 3 on Fig 5); E, F–stromatoporoid detrital limestones (rudstones) with densely packed, grain-supported texture composed of stromatoporoid fragments (allobiostrome, sample 2 on Fig 5); G, H–stromatoporoid detrital limestones (grainstones and rudstones) with stromatoporoid fragments, *Amphipora*, and shell debris (parabiostrome, sample 4 on Fig 5); all scale bars are 5 mm long.

### Morphometrical and taphonomical features of stromatoporoid skeletons

Stromatoporoids representing three types of occurrences were studied in the Ołowianka Quarry succession (Fig 5) – (i) an allobiostromal accumulation of redeposited and reworked material (A), (ii) a parabiostromal accumulation of partly reworked material (P), and (iii) *in situ* and redeposited skeletons from all other beds. In combination with taphonomical and sedimentological data, the results of morphometrical analysis were interpreted in terms of the stromatoporoid growth environments and the sedimentary conditions that led to their accumulation.

**Allobiostrome.** The stromatoporoid assemblage accumulated in the allobiostrome shows several characteristic morphometrical features that discriminates it from the other studied assemblages (Table 1; Fig 8A–8D). The shapes, and other attributes, of 273 stromatoporoid skeletons from the allobiostrome were well-enough preserved to be taken into consideration

**Table 1. Basic morphometrical and taphonomical features of the studied stromatoporoids.**

| | Allobiostrome (A) | Parabiostrome (P) | Remaining beds |
|---|---|---|---|
| **Total number** | 273 | 69 | 37 |
| **Shape profile** | | | |
| Laminar | 7 (3%) | 8 (12%) | 16 (43%) |
| Low domical | 33 (12%) | 17 (25%) | 12 (32%) |
| High domical | 92 (34%) | 20 (29%) | 8 (22%) |
| Bulbous | 141 (52%) | 24 (35%) | 1 (3%) |
| **Initial surface** | | | |
| Flat | 14 (5%) | 34 (49%) | 17 (46%) |
| Initial elevation | 74 (27%) | 32 (46%) | 6 (16%) |
| Anchor | 62 (23%) | 3 (5%) | 14 (38%) |
| Encrusting | 0 | 0 | 0 |
| Undetectable | 123 (45%) | | |
| **Surface character** | | | |
| Smooth | 267 (98%) | 48 (69%) | 11 (30%) |
| Ragged | 6 (2%) | 21 (31%) | 26 (70%) |
| **Latilaminae arrangement** | | | |
| Enveloping | 210 (77%) | 44 (64%) | 8 (22%) |
| Non-enveloping | 39 (14%) | 25 (36%) | 29 (78%) |
| Undetectable | 24 (9%) | | |
| **Redeposition/orientation** | | | |
| Growth orientation | 39 (14%) | 5 (8%) | 28 (76%) |
| Overturned | 224 (82%) | 61 (88%) | 9 (24%) |
| Undetermined | 10 (4%) | 3 (4%) | |
| **Preservation** | | | |
| Complete | 273 | 69 | 37 |
| Parts missing or broken (not included in analysis) | 94 | 51 | 2 |

in the present study. Only specimens considered to be complete (or almost complete) were subsequently studied, which constitutes around 75% of all the skeletons exposed in the allobiostrome (Table 1). A limited proportion of stromatoporoids (less than 15%) were found in a normal upward orientation, which may suggest their preservation in an *in situ* growth position (although this is not necessarily true, especially in the case of equidimensional specimens). All other specimens were either visibly overturned and thus considered to be redeposited, or their orientation could not be unambiguously determined.

Perhaps the most conspicuous morphometric feature of the allobiostrome stromatoporoids is the distinct domination of high-profile specimens. Over 85% of studied skeletons (233 specimens) possessed bulbous and high-domical shapes (141 and 92, respectively). Stromatoporoids with bulbous shapes, alone, constitute more than half of the studied assemblage. The most typical shape is a roughly equant equidimensional body, which in cross-section is usually traced as a circle or as a compacted ellipse (Fig 8A–8D). Flat, oblate forms also occur; however, these are much less abundant, especially those forms with a laminar shape (Table 1).

Subtle weathering of the stromatoporoids exposed on the quarry walls delineated latilaminae arrangements within most skeletons (Fig 8). The vast majority of specimens exhibit enveloping latilaminae arrangements (84% among specimens in which latilaminae arrangements could be determined). This suggests that, in the preponderance of cases, the high shape profile describes not only the whole skeleton, but also the growth form (i.e., the living surface profile),

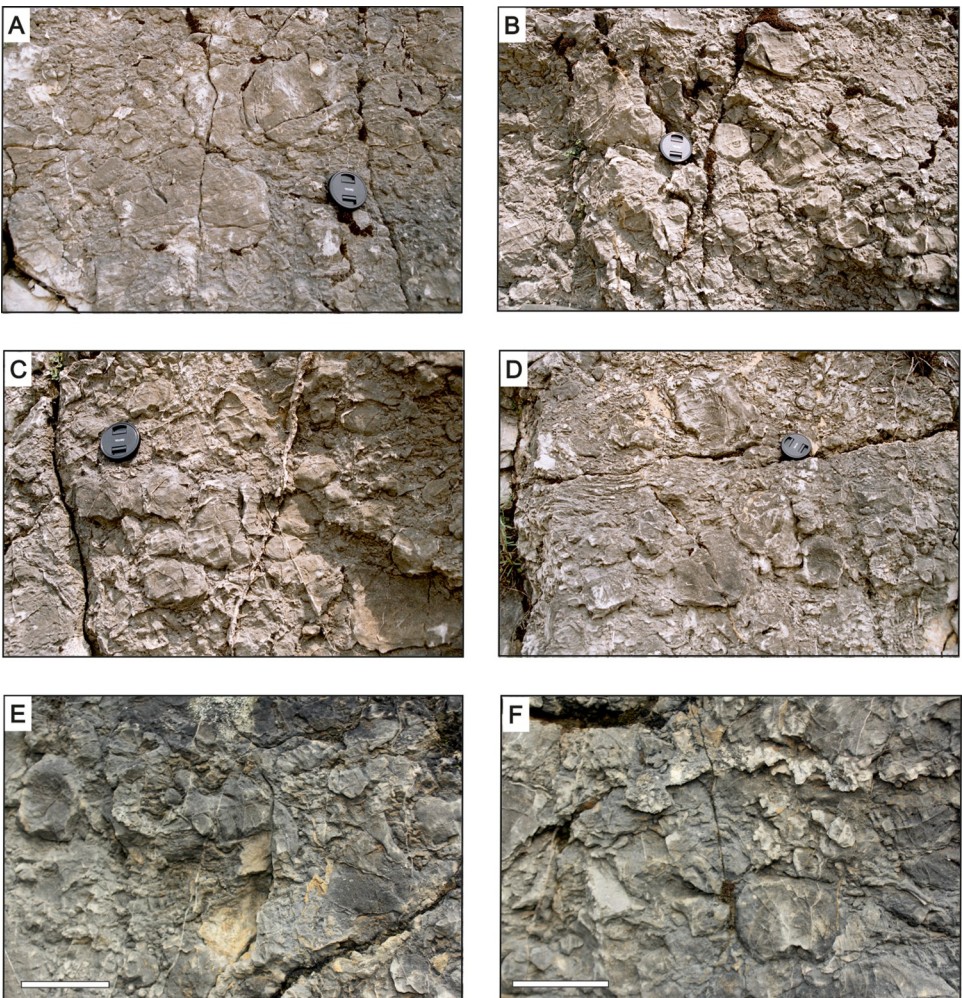

**Fig 8. Weathered surfaces with salient stromatoporoid skeletons.** A-D–allobiostrome (A); E, F–parabiostrome (P). Scale bars on E and F are 10 cm long.

which reflects the part of the skeleton protruding from the sediment above the seafloor during the final stage of stromatoporoid growth. To an even greater extent, stromatoporoids in the allobiostrome are dominated by forms with a smooth upper surface (and sides); ragged varieties are almost completely absent. Only six specimens were classified as ragged; however, it must be noted that there is the possibility that some studied stromatoporoids were potentially preserved incompletely, but this partial preservation was not detected–that is, they may have initially had protrusions that broke off. In such case, their skeletal surface would have been erroneously classified as smooth.

With regards to initial surfaces, an initial elevation morphology is the most abundant, while anchor morphologies are almost as common (Table 1). Not all sections through the stromatoporoid skeletons allowed explicit determination of this attribute; consequently, more than a third of specimens were left unclassified in this regard. Nonetheless, among those studied, around 50% of skeletons have an initial elevation, a further 40% are characterized by an anchor, whereas flat lower surfaces–which elsewhere are dominant through the Ołowianka succession–are very rare, confined to only laminar and some low domical forms. No encrusting forms were recognised.

In the allobiostrome, numerous skeletons that are either visibly fragmented and/or broken, or otherwise apparently not complete, are present (Table 1). These specimens were generally not considered in the present study; however, some of their attributes were also observable. If recognizable, the fragmented specimens generally reveal the same dominant features as their complete counterparts in all morphological characteristics, including their shapes, initial surfaces, latilaminae arrangements, and surface characters.

As no detailed dimensional measurements were made of the studied stromatoporoids, the sizes of individual specimens were not determined; that being said, some qualitative elucidation of size distribution is possible. The largest complete specimens in the allobiostrome have a diameter (or, rather, longest dimension seen in cross section) between 30–40 cm; however, a more typical value lies somewhere around 20 cm. An assortment of smaller specimens is also present. What is characteristic, however, is that no distinct morphometrical differences between specimens of different sizes have been noticed – that is, there is no growth allometry, which otherwise is typical amongst stromatoporoids. In addition, the dominance of other morphological attributes is unrelated to skeletal size; a partial exception to this generality may be the latilaminae arrangements, which more often tend to be non-enveloping for the larger skeletons relative to smaller stromatoporoids.

In summary, the most typical stromatoporoid morphotype from the allobiostrome is a redeposited bulbous or high domical form with a smooth surface, an enveloping latilaminae arrangement, and an initial elevation.

**Parabiostrome.** A total of 69 stromatoporoid skeletons were analysed from a parabiostrome outcropping in the upper part of the exposed succession (Figs 6E–6F and 8E–8F). The studied stromatoporoid assemblage is distinctly different than that of the allobiostrome, at least in terms of stromatoporoid morphology (Table 1).

The distribution of stromatoporoid shapes in the parabiostrome assemblage is more even: that is, there is not the same degree of high shape profile dominance as in the allobiostrome (Table 1), although the high domical and bulbous forms combined still constitute 64% of the studied group. Additionally, the distribution of latilaminae arrangements and upper surfaces is more balanced, although the enveloping and smooth varieties, respectively, are still more common. Among the styles of initial surfaces, flat is the most common, closely followed by initial elevation, which is markedly different relative to the allobiostrome assemblage.

Due to the greater proportion of characteristic, non-equidimensional shapes (such as laminar and domical), skeletal orientation could be determined for almost all specimens (Table 1). Most skeletons are overturned or tilted, with only five low shape specimens preserved in an upward growth position, which may suggest, albeit not necessarily, that they were found *in situ*. The latilaminae arrangements of some skeletons indicate that they were at least partially overturned, and subsequently continued their growth along a new axis. A considerable proportion of the skeletons–almost half–are incompletely preserved; a greater proportion than in the allobiostrome.

The size range of parabiostromal stromatoporoids is generally comparable to those of the allobiostrome: the largest specimens attain a largest dimension of over 30 cm, while a typical largest dimension size is around 20 cm. However, in this assemblage, a distinct relation between skeletal capacity and shape can be observed. The largest specimens are those characterised by relatively low shape profiles – laminar and low domical – whereas bulbous forms never exceed a diameter of 20 cm (Fig 8E–8F). Moreover, the morphological characteristics of fragmented and incomplete stromatoporoids apparently differ from those preserved as complete skeletons, with the former dominated by bulbous forms with primarily enveloping latilaminae–yet another point of contrast to the allobiostrome.

**Non-biostromal beds.** Single stromatoporoids, loosely scattered in the sediments rather than forming skeletal accumulations, are present in several beds in the Ołowianka Quarry, particularly in the upper part of the succession (Fig 5). Although the group is stratigraphically artificial and does not represent any particular sedimentary event, their morphometric features have also been studied here for comparison with the biostromal event bed associations. A total of 37 specimens was exposed in proper cross-sections and preserved in a way that enabled the identification of the main macroscopic morphometric features, such as initial surfaces or latilaminae arrangements: only these specimens were taken into consideration.

The stromatoporoid 'assemblage' in non-biostromal beds is, in many aspects, strikingly different than the biostromal assemblages (Table 1). Firstly, it is mainly represented by *in situ* specimens. Some skeletons are tilted, which may suggest some movement: however, no overturned and or broken/fragmented forms are observed, with only minor exception. Secondly, it is dominated by specimens with non-enveloping latilaminae arrangements and ragged surfaces, the opposite to the patterns recognized in both biostromal accumulations. Finally, the most typical shapes are laminar and low-domical forms with a flat initial surface or an anchor.

## Internal sedimentary structure of stromatoporoid beds

The studied stromatoporoid allobiostrome and parabiostrome differ distinctly with regards to their internal structure, including features such as stromatoporoid packing, sorting, and distribution.

**Allobiostrome.** In the quarry, the lateral distribution of accessible outcrops that expose the main allobiostrome is limited to some tens of meters, which, however, is sufficient to establish that its thickness is variable and dynamic over small distances. The maximum thickness reaches 6 meters: however, in the most accessible locality, where most of the detailed sedimentological and morpho-/taphonomical observations were made, it ranges between 3.5 and 4 meters (Fig 6A), as illustrated on Fig 5. The lower boundary of the bed is clearly erosional. The allobiostrome rests on a thick pelitic limestone bed with scarce *Amphipora* and isolated massive stromatoporoid skeletons, typical of the Stringocephalus Beds. Its upper boundary is also abrupt, and the allobiostrome is overlain by similar facies as those below. The allobiostromal bed is massive, without any internal layering or banding.

The allobiostrome is mainly composed of massive stromatoporoids, predominantly overturned and thus not representing *in situ* growth positions. Complete skeletons co-occur with fragmented specimens. Massive stromatoporoids are accompanied by abundant debris derived from dendroid forms – primarily *Amphipora* and *Stachyodes*–as well as occasional tabulate and rugose coral fragments. The skeletons are densely packed, and through most of the outcrop form the basis of a clast-supported texture (Fig 9A): there is only a small amount of micritic matrix containing scattered brachiopod and ostracod shell debris.

The internal structure of the allobiostrome is nearly uniform; no distinct vertical differentiation is observed within the bed. The material is unsorted, with stromatoporoids of all size classes and other bioclasts co-occurring. Perhaps the only vertical differentiation is the overrepresentation of the largest stromatoporoid skeletons in the lowest meter of the bed. Additionally, no vertical sorting is observed with regards to stromatoporoid shapes and other external features, as well as with their preservation state.

**Parabiostrome.** The thickness of the parabiostrome varies between 1.3 and 1.5 meters. Both the upper and lower boundaries are sharp and abrupt: the lower is distinctly erosional. The bed rests on a thick layer of pelitic, unfossiliferous limestone with scarce *Amphipora*, and is capped by pelitic limestone with isolated stromatoporoids and *Amphipora*.

The parabiostrome is mainly composed of massive stromatoporoids, which are accompanied by fragmentary tabulate and rugose corals, broken *Amphipora* branches, and brachiopod

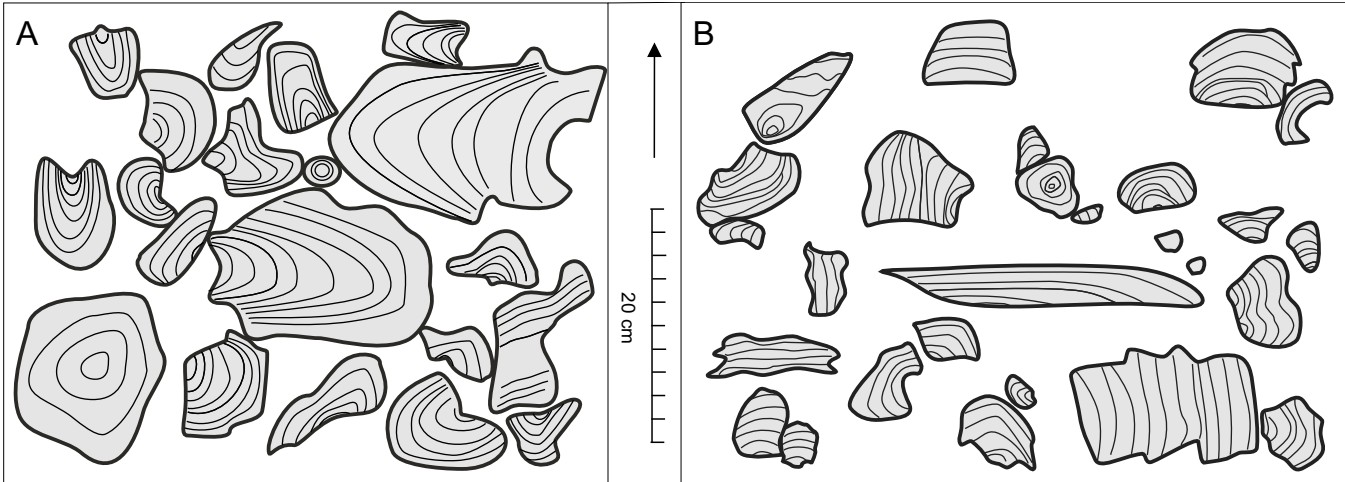

**Fig 9. Sketches of stromatoporoid assemblages visible on weathered rock surfaces.** A–allobiostrome fragment representing a typical clast-supported stromatoporoid accumulation dominated by high-domical and bulbous forms with smooth surfaces and enveloping latilaminae arrangements, and with initial surfaces dominated by initial elevations and anchors; B–parabiostrome fragment representing a typical scattered accumulation, consisting of mixed complete and fragmented stromatoporoid specimens with a wide variety of shapes, surface characters, and latilaminae arrangements (middle/upper part of the parabiostrome).

and ostracod shell debris. In stark contrast to the allobiostrome, the composition and internal structure of the parabiostrome perceptibly changes along a vertical profile. The bottom third of the bed generally exhibits a clast-supported texture, densely packed with stromatoporoid skeletons; in the upper portion of the parabiostrome, the bioclasts density gradually decreases (Fig 9B). Moreover, there is a degree of graded bedding with respect to average bioclast size. In general, larger specimens are proportionally more abundant in the lower part of the bed; however, individual large skeletons can be also found in the upper portion. The proportion of complete and fragmented specimens also varies vertically, as broken fragments are relatively more plentiful in the lower part of the bed. In general, broken fragments are much more abundant in the parabiostrome compared to the allobiostrome.

Another vertically variable feature in the parabiostrome, again in distinct contrast to the allobiostrome, is the shape segregation of stromatoporoids through a vertical section. Large skeletons with low shape profiles and flat lower surfaces are easier to find in the lower part of the biostrome, whereas bulbous and high-domical specimens with initial elevations are dominant in the upper parabiostrome.

## Discussion

In the Devonian Holy Cross carbonate platform, Givetian to Frasnian shallow-water facies, largely representative of calm settings, are interbedded and/or fringed upon by various beds which, collectively, point to the intermittent influence of high-energy sedimentary events on the local depositional record. Most commonly, these are represented as various coarse-grained deposits including calcarenites, carbonate breccias, bioclastic beds, and flat pebble conglomerates, usually interpreted as storm- [38–40] or gravitational associated deposits [41, 42]. However, many similar deposits could also have resulted from tsunamis and associated phenomena, as has been demonstrated in the Silurian of Podolia [30, 32, 35, 36, 80]. The possibility of palaeotsunamis in the Holy Cross Mountain Devonian has been previously suggested by Kaźmierczak and Goldring [37]. Conceivably, tsunamis could have been triggered by local phenomena, such as earthquakes and submarine mass movements associated with

syndepositional tectonic activity, particularly along the northern and southern margins of the shallow water carbonate platform in the central (Kielce) region of the Holy Cross area [7, 10]. This activity is documented, for example, by the occurrence of multiple generations of neptunian dykes, intraformational breccias, zebra rocks, and so forth penetrating more typical Kowala Formation deposits in a number of settings [81], and other independent evidence of synsedimentary tectonics [7, 14]. However, considering the scale of modern tsunami events (e.g., the 2004 Sumatra tsunami, which caused massive damage across the periphery of the Indian Ocean), a potential tsunami source may have been located in a more distant region of the Variscan realm; if so, waves could travel across oceanic basins and epicratonic seas, rapidly growing in height and strength upon reaching the shallow waters of the Holy Cross carbonate platform. The Ołowianka Quarry is located close to the southern margin of this platform, which lends credence to such an proposition.

Sedimentological analysis of two characteristic biostromal accumulations (an allobiostrome and a parabiostrome) in the Ołowianka Quarry, combined with morphometrical and taphonomical studies of the stromatoporoid skeletons from which the biostromes are composed, were used to assess the character of the high-energy sedimentary phenomena responsible for their accumulation. As mentioned above, selected features of the stromatoporoid beds can be interpreted both with regards to the original stromatoporoid habitats and the high-energy processes leading to their transport and redeposition. Clear differentiation between the two beds suggest that the thick allobiostromal bed ultimately arose from a tsunami, while the parabiostrome is a more typical storm tempestite; Various methods of distinguishing tsunamites from tempestites in the geological record have been discussed by many authors [82–91], including in studies of modern examples [92–97]. This task has proven to be complicated, and according to some authors, utterly impenetrable. Here, I present characteristic morphometric, taphonomical, and sedimentary features of stromatoporoids and their accumulations that can be used to discern the two cases, at least in situations where stromatoporoids are present.

## Original stromatoporoid growth environments

**Allobiostrome.** All morphometrical features of the stromatoporoids composing the allobiostrome reflect relatively calm sedimentary conditions in their original growth environment. The dominance of high-shape profiles with prevailing enveloping latilaminae arrangements are suggestive of slow, stable depositional rates [68]. Low burial ratios (as indicated by enveloping latilaminae) and the general lack of sediment increments within the skeleton (i.e., the absence of ragged varieties) indicate that the stromatoporoids stood high above the sediment surface with their skeletons exposed, which rendered them very vulnerable to redeposition. In light of this susceptibility to transport, the observation that many skeletons attained relatively large dimensions suggests that calm conditions prevailed over periods long enough for large stromatoporoids to grow undisturbed. If the latilaminar structure of stromatoporoids reflects annual banding, as has been previously suggested [71], this suggests persistent calm episodes lasted hundreds of years. The stability of growth conditions is also supported by the absence of an allometric shape change tendency linked with skeletal growth in the studied assemblage [61], which otherwise is a common feature among stromatoporoids [66, 69].

In general, the most common combination of stromatoporoid morphometric features is a high shape profile (predominantly bulbous), smooth upper surface, and a non-enveloping latilaminae arrangement [54]. Such forms are routinely interpreted to represent calm, usually restricted shallow-water settings with muddy carbonate bottoms, which are typical stromatoporoid habitats [52, 65]. The dominance of non-enveloping varieties, which indicates that growth was restricted to only the upper part of the skeleton, combined with the lack of

sediment increments, suggestive that there were no rapid sediment inputs, is interpreted as the result of growth in settings with a cloudy turbid bottom water layer [61]. Such a layer could have been generated by the agitation of fine sediment particles during recurrent turbid events, such as storms. After settling from suspension, these particles would have clogged the tiny pores of the stromatoporoid, inhibiting soft tissue growth on the lower parts of the skeletal surface. However, the low proportion of non-enveloping varieties in the allobiostrome assemblage suggests that, in this case, the water column was clear with no bottom turbid layer, and furthermore that pore clogging was not an routine hazard. A similar conclusion can be drawn from the relatively common occurrence of initial surfaces developed as anchors. In environments with cloudy, turbid bottom waters, small elevations are preferentially inhabited by stromatoporoids, which results in the dominance of initial elevation or encrusting initial surfaces [24, 61, 68].

In combination, these stromatoporoid features indicate that the stromatoporoids comprising the allobiostrome assemblage grew undisturbed through their entire life history (lasting up to hundreds of years) in a clear water setting. It is likely that such environments were located below storm wave base, and therefore were not interrupted by recurrent high-energy events, such as storms. However, the large specimen dimensions indicate that local conditions must have been suitable for efficient skeletal construction, which would have only been possible at depths with sufficient light [17, 18, 53, 72]. The outer shelf, which was located far from the fine terrigenous input sourced from the emergent, weathering portions of the carbonate platform, is the most feasible setting for which this combination of environmental conditions could occur. In this framework, the most feasible driver of mass redeposition of stromatoporoids from the outer shelf was a tsunami. It is worth noting that tsunami waves travelling through open-marine areas are usually devoid of much suspended sediment until they reach the coastal zone [89].

**Parabiostrome.** As in the case of the allobiostrome, the parabiostrome stromatoporoid assemblage is mostly composed of redeposited and/or overturned skeletons. Even those few specimens labelled as "in growth position" may have been subjected to transport and simply redeposited with the original skeletal orientation. This indicates that, as in the allobiostrome, the stromatoporoid morphometrical features reflect their original growth habitats rather than their final accommodation place.

The diversity of stromatoporoid shapes and other external features in the parabiostrome is much more disparate than in the allobiostrome, which suggests that their original growth environment must have been more dynamic and variable, or alternatively that particular specimens were derived from various sources and thus represent various habitats [52, 57, 61, 66]. Skeletons occur across the whole range of typical stromatoporoid shapes (Table 1): high shape profiles are the most common, but not to the same degree as in the allobiostrome. Perhaps even more variable than the whole skeletal shapes were the living surface profiles, which reflect the final growth forms protruding over the sediment surface. A substantial proportion of the specimens exhibit non-enveloping latilaminae arrangements and ragged upper surfaces (Table 1), indicative of a high burial ratio and a discrepancy between the shape and growth form. Such varieties most commonly occur among both low and high domical forms. Another feature that distinguishes the parabiostrome stromatoporoid assemblage from its allobiostromal counterpart is the relative proportions of various initial surface styles. In the parabiostrome assemblage, flat surfaces and initial elevations are almost equally abundant, accompanied by only a few specimens with anchors.

As a result of these morphologic combinations, stromatoporoid forms in the parabiostrome are either derived from various growth environments or represent a setting that itself is internally diverse. Regardless, however, the original growth habitats were subjected to recurrent

periods of elevated water column turbidity connected with rapid sediment influx, as reflected in the relatively common occurrence of ragged varieties, with sediment increments disrupting skeletal growth [52, 69, 71, 75]. The roughly comparable abundances of ragged surfaces and non-enveloping latilaminae arrangements suggest that it was not the clogging of pores in cloudy bottom waters by tiny particles that interrupted growth, but rather partial burying by coarser material [61, 77]. Furthermore, a high proportion of forms had initial elevations, suggesting that minor elevations were preferably inhabited and pointing towards the considerable hazard of sediment burial during the early stages of skeletal growth [24, 68].

The shapes of stromatoporoids from the parabiostrome are size dependant, in contrast to those in the allobiostrome. This is yet another indication that the original growth conditions were not as stable as in the allobiostrome, and that calm periods were repeatedly interrupted by high-energy episodes. Specimens characterised by high profiles and low burial ratios, and thus susceptible to redeposition, did not attain large dimensions. Larger stromatoporoids are mainly represented by laminar and low domical growth forms, which, if they managed to avoid sediment burial in their early growth stages, were better anchored in the sediment and thus less vulnerable to transport. As a point of comparison, among the incomplete fragments bulbous forms with enveloping latilaminae are dominant; this probably results from multiple cycles of redeposition.

All of these stromatoporoid features point towards an original growth environment with dynamic and variable sedimentary conditions. Shallow settings above storm wave base, within reach of fine-grained sediments derived from emergent areas, are the most probable localities. Extensive shoals with small, patchy underwater elevations, inhabited by massive stromatoporoids, amphiporids, and tabulates correspond well to this picture, and are characteristic of the Stringocephalus and Sitkówka Beds [9, 79]. In such settings, basic environmental factors, such as water turbulence, sedimentation rate, and bottom water clearness could vary over short distances, which is reflected in the disparity and variability of the observed stromatoporoid morphometric features.

The parabiostrome stromatoporoid assemblage differs in some important respects from the *in situ* stromatoporoids of the non-biostromal beds exposed in the quarry, which represent shallow water lagoonal settings. The latter are dominated by low profile forms with non-enveloping latilaminae, ragged surfaces, and flat initial surfaces and anchors. These are the features indicative of growth in an intermittently turbulent environment; however, they also characterize specimens well-adapted to resist exhumation and redeposition. Those forms more susceptible to transport, such as bulbous forms with enveloping latilaminae, are almost totally absent in the non-biostromal beds, although they are common in the parabiostrome. This suggests that the skeletons accumulated in the parabiostrome may have been derived from similar environments *via* the preferential exhumation and redeposition of more labile forms, of which the non-biostromal beds are depleted [30].

## Stromatoporoid transport and bed accumulation

**Allobiostrome.**   The landward transport and redeposition of open marine macrofauna (and microfauna) characteristic of the middle and outer shelf by tsunami waves is a well-recognised phenomenon [84, 98]. Additionally, these processes have been reported in the context of Upper Devonian flat pebble conglomerates of an alleged tsunami origin [37]. A tsunamic interpretation has also been presented for stromatoporoid accumulations and conglomerates in lagoonal successions of the Silurian in Podolia [30, 35, 36]. Some of the features observed there, and interpreted as pointing to a tsunami origin, also occur in the Ołowianka allobiostrome.

Due to the very long periods of tsunami waves, water movement occurs throughout the whole water column, reaching the seabed even at great depths [80, 91, 99]; it is therefore capable of setting sediments at depths well below storm wave base in motion [99–102]. Moreover, sea level drop between successive tsunami waves can lead to the emergence of vast areas subsequently prone to erosion [93, 103]. Reef elements, and other bioaccumulations forming and/or associated with bottom elevations, are the most common sources of redeposited material [89, 98]. Modern tsunami-derived boulders largely consist of light, porous, reef limestones, which in some cases attain very large dimensions [98, 103, 104].

Palaeozoic stromatoporoids usually did not construct rigid frameworks, but rather grew as distinct, individual specimens [55, 56, 64, 66]. This, indeed, is the dominant growth mode of all known Devonian stromatoporoids from the Holy Cross Mountains [18, 19, 21, 24]; no large reef structures are present in the area [9, 14]. Microbially bound mounds with stromatoporoids and corals from the Frasnian are the most prominent biohermal structures [15, 16, 18], although there are some indications that rigid reefs may have been present [17]. In the case of the allobiostrome stromatoporoids, the separate growth of individual specimens is confirmed by the complete absence of encrusting initial surfaces and overgrown skeletons. No multidome clusters, formed by the coalescence of closely growing skeletons [24], have been found.

Erect stromatoporoids growing as solitary individuals in soft sediments with their entire skeletons protruding over the seafloor surface, such as those accumulated in the allobiostrome, were particularly vulnerable to redeposition. This vulnerability was further enhanced by their low specific weight, which augmented buoyancy [58]; in most cases, prior to the post-mortem crystallization of sparry calcite the internal voids of stromatoporoid skeletons were probably filled with water [59]. The roughly equant, equidimensional stromatoporoid skeletons, represented by the high-domical and bulbous forms, were especially susceptible to long-distance transport [54, 57]. Flow velocities were clearly fast enough to carry all stromatoporoid skeletons in the same manner regardless of size, as indicated by the co-occurrence of large and small specimens. It must be acknowledged the largest stromatoporoids had diameters of tens of centimetres, which is substantially smaller the typical size of modern *tsunami-ishi* boulders [104–109]. However, tsunamites are source dependant, like any other detrital sedimentation, and as such can only incorporate the elements present in the source areas [83, 84, 87, 89, 90, 99, 103]. The absence of large boulders is therefore not a substantial counterargument to a posited tsunami origin; on the contrary, even large catastrophic events, such as the 2011 Tohoku tsunami, commonly leave little sedimentary imprint and are represented by single layers of coarse-grained material or redeposited mud [101]. In this case, individual stromatoporoid skeletons were simply the largest elements available for redeposition. It is worth noting that the absence of lateral size segregation is another characteristic feature of tsunami deposits that distinguishes them from tempestites [106, 107].

Tsunami waves can transport material in various ways–via traction, suspension, and saltation [91, 103, 110–114]. In the case of the Ołowianka allobiostrome, most skeletons were probably carried by a combination of traction and suspension mechanisms. The role of suspension transport is suggested by the relatively low admixture of broken fragments (Table 1) and the high buoyancy of stromatoporoid skeletons, and may be responsible for onshore redeposition over longer distances. On the other hand, the dominance of traction in the final depositional episode is indicated by clast supported textures and the lack of vertical size sorting [36].

The thick allobiostrome is developed as a massive body without internal banding or layering, indicative of a single act of deposition. Studies of contemporary tsunamis reveal that the entire process usually embraces a series of alternating erosional and depositional episodes associated with individual waves and their backwash flows, both in onshore and shallow

offshore settings [90, 110]. However, in the fossil record–especially in shallow water settings–the most recently deposited material is often eroded and removed by subsequent waves [115]. The variable thickness and sharp, erosional boundaries of the allobiostrome suggests such a scenario is applicable to this case. Moreover, in areas characterised by a very gentle gradient, as in the studied setting, the backwash flow is usually insignificant [105]. During the deposition of the Stringoce-phalus Beds, the Ołowianka region was located on the southern slope of a bank complex, flanked from the north and south by relatively deeper intrashelf basins [9, 14] (Fig 2B). No large land-masses were present in the vicinity, and consequently there may have been no significant back-wash flow at all; this would explains the lack of characteristic tsunamite features such as an admixture of offshore transported terrestrial material [114]. In this case, a single sedimentary event, possibly associated with a drop of flow speed and transport power in the last wave of the tsunami wave train, was responsible for the deposition of the thick allobiostromal layer.

**Parabiostrome.** For shallow water carbonate platform settings–especially for areas located close to the platform edges and facing deeper, open marine waters–the occurrence of high-energy beds in otherwise calm bank and lagoonal facies is most naturally explained via a tem-pestite mechanism. Indeed, such interpretations have been proposed for various high-energy facies that interbed, and fridge the flanks, of Devonian shallow-water carbonate platform deposits of the Holy Cross Mountains [38–40]. The average recurrence intervals of even partic-ularly fierce storms are at least an order of magnitude shorter than for tsunamis [84, 94, 110]. Therefore, in the absence of directly contradictory features (Table 2), a storm interpretation is usually the preferred, and default, explanation [30, 32, 35], as in the parabiostrome.

The parabiostromal stromatoporoid accumulation differs in several ways from its allobios-tromal counterpart, and lacks the critical features that point to a tsunami origin of the latter. Most importantly, the original growth locality of the redeposited stromatoporoids–that is, the source area of both the skeletal and non-skeletal material laid in the bed–was in a periodically turbulent zone, as indicated by the morphometrical features of the stromatoporoid skeletons. Such a habitat would have been repeatedly subjected to both interruption of stromatoporoid growth and the elimination of forms susceptible to redeposition. This original environment most easily can be interpreted as being above storm wave base, with recurrent storm activity.

Another feature distinguishing the two stromatoporoid accumulations is the internal vari-ability of the parabiostrome across a vertical profile, in contrast to the homogenous, massive structure of the allobiostrome. The parabiostrome cannot be treated as a single act of deposi-tion, even though it is thinner than the allobiostrome. On the contrary, various elements point to different transport and deposition mechanisms.

Graded bedding, as demonstrated by vertical stromatoporoid size sorting and an upwards decrease of skeletal material content within the bed, points towards an important role of depo-sition from suspension. Round, well preserved, equidimensional bulbous and high domical forms of limited size, which occur mostly in the upper part of the bed, represent the skeletons most susceptible to redeposition and transport [54, 57]. Indeed, these were the specimens most easily transported in suspension over longer distances, in a comparable manner as in the allobiostrome. In contrast, however, some specimens, especially the large laminar and low domical forms present in the bottom part of the bed, are typically tilted but not overturned, which suggests that only very limited transport occurred [25, 54]. Other forms are clearly over-turned. They are commonly accompanied by broken and fragmented skeletons, which points to repeated exposure to high-energy conditions [53]. Additionally, several specimens occur that continued their growth after redeposition. In conjunction, these features point to a multi-phase process of transport and deposition, mostly via traction.

The differential distribution and spatial separation of the stromatoporoid skeletons by dimension and shape within the parabiostrome, suggestive that various forms were subject to

**Table 2. Main stromatoporoid morphometrical and taphonomical features, sedimentary attributes of stromatoporoid beds, and their interpretation in terms of original stromatoporoid growth environments, transport mode of redeposited material, and character of final accumulation.** T – attributes typical and/or expected for tsunamites, T – attributes unexpected for tsunamites, S–attributes typical and/or expected for storm deposits (tempestites), S – attributes unexpected for storm deposits.

| Observed features | Stromatoporoid growth environment | Transport and final accumulation of redeposited material |
|---|---|---|
| **Allobiostrome** | | |
| High profile shapes with smooth surfaces and low burial ratio | Calm environment with low depositional rate (S, T)) | Very high susceptibility to redeposition (S) |
| No size allometry, large dimensions | Stable growth conditions (S, T) | ———————— |
| Anchors, enveloping latilaminae | Clear waters, absence of cloudy turbid bottom layer (T, S) | ———————— |
| Small admixture of broken fragments | ———————— | Single act of deposition, transport mainly via suspension (T, S) |
| Thick massive bed without internal layering or banding | ———————— | Single act of deposition (T, S) |
| Sharp boundaries, erosional lower contact, variable thickness | ———————— | High energy sedimentary environment (T, S) |
| Lack of graded bedding and size sorting | ———————— | Deposition mainly from traction, rapid single act of deposition (T, S) |
| Clast supported textures | ———————— | High energy sedimentary environment (T, S) |
| **Parabiostrome** | | |
| Diversity of stromatoporoid shapes and other external features | Dynamic and variable and/or diversified growth environment (S) | ———————— |
| Ragged surfaces, non-enveloping latilaminae, initial elevations | Periods of rapid sediment influx interrupting growth (S) | ———————— |
| Size allometry–lack of high profiles among larger forms | Elimination of specimens susceptible to redeposition (S, T) | Multiple episodes of redeposition (S, T) |
| Sharp boundaries, erosional lower contact, variable thickness | ———————— | High energy sedimentary environment (T, S) |
| Size and shape sorting, graded bedding | ———————— | Various means of transport–traction, saltation, suspension (S) |
| Clast supported textures | ———————— | High energy sedimentary environment (T, S) |

different modes and distances of transport, indicates that the transportation forces were not as powerful as in the allobiostrome. The round forms were easily transported over longer distances; by contrast, flat stromatoporoids, which were better rooted and less susceptible to exhumation, and those with ragged margins were moved over only limited distances (but, most likely, multiple times). During tsunami-induced redeposition, no such differentiation is expected, as the tsunami wave can readily carry very large and heavy elements without imparting size segregation [106].

Some features common for both the allo- and the parabiostrome (Table 2), such as the erosional character of the lower and upper boundaries, lateral thickness variability, and clast-supported textures in at least parts of the beds, are characteristic for both tsunamites and tempestites [83, 84, 91, 92, 116]. In this case, a storm interpretation seems the most probable.

## Record of high-energy sedimentary events in the Kowala Formation

The Middle and Upper Devonian Kowala Formation, which reaches up to 800 m thickness and forms the bulk of the epicontinental shallow-water carbonate sequence of the Holy Cross area [9, 20], abounds in stromatoporoid-coral beds of various character [14, 15, 21]. Many of these beds are composed predominantly or even solely of redeposited stromatoporoid skeletons, although autobiostromes and bioherms are also common. In most cases, the genesis of

redeposited beds can be attributed to storm-influenced deposition in shallow-water areas. However, for at least some beds a tempestitic origin cannot be upheld. One such example is presented in the present study, where sedimentological, taphonomical, and morphological arguments point towards the tsunamite nature of a thick bed composed of redeposited stromatoporoid skeletons.

Erosion in deep-water environments, large-scale onshore transport of coarse, unsorted detrital material, and associations with other evidence of seismic activity are among the most common attributes linked with a tsunami origin of high-energy deposits [84, 90, 92, 93]. The first two are manifest in the allobiostrome. The influence of syndepositional tectonic activity on the development of the Kowala Formation and contemporary equivalents–which must have been associated with seismic events that may have had the capacity to induce tsunamis–is widely acknowledged [7, 14, 42].

In most cases, the unequivocal recognition of palaeotsunamites, and in particular discerning them from more typical palaeotempestites, is an extremely difficult task. This is largely caused by the common absence of characteristic features and the general similarity of the depositional record of abrupt, high-energy marine processes [83, 90, 91]. Consequently, tsunamites are rarely identified in the sedimentary record, and the influence of tsunami induced phenomena on deposition is commonly underestimated [117]. In this context, studies of variously developed stromatoporoid beds, and particularly the analysis of the stromatoporoid morphometric features, provide a unique opportunity to identify palaeotsunamites in the geologic record [30, 32, 36]. Stromatoporoid facies are common among shallow water carbonate deposits from the Middle Ordovician to the Upper Devonian (Frasnian) [49, 54] and in this timespan they may serve as potential indicators of palaeotsunamites.

Further detailed studies are needed to determine the potential tsunamite nature of other stromatoporoid accumulations in the Kowala Formation; however, it seems feasible that at least some of them share the features invoked here as pointing to tsunami-induced deposition.

## Conclusions

The calm, shallow water Stringocephalus Beds exposed in the Ołowianka Quarry are punctuated by high-energy facies composed of redeposited massive stromatoporoid skeletons. Analysis of a thick allobiostromal and a thinner parabiostromal stromatoporoid accumulation is suggestive of a tsunami origin of the former, and a storm origin of the latter. These interpretations are based on the following observations and arguments: (i) Morphometrical features of stromatoporoids from the allobiostrome and from the parabiostrome differ from one another, pointing to different original habitats of the redeposited specimens; (ii) In the allobiostromal assemblage, the original stromatoporoid habitat was located in a calm setting below storm wave base in an environment undisturbed by high energy episodes, allowing a long, undisturbed growth history of massive stromatoporoids. The erect high-profile forms that dominate this assemblage were very susceptible to redeposition, yet commonly attained large dimensions. The large scale and mass onshore redeposition of stromatoporoid skeletons from this type of setting was possible only due to an extraordinary sedimentary event that caused erosion and transport at considerable depths: a tsunami is the most reasonable explanation. The growth environment of the parabiostrome specimens was much more diversified, and punctuated by high-energy episodes that interrupted skeletal growth. Redeposition of the stromatoporoids accumulated in the studied parabiostrome did not require special phenomena, and most probably took place during storms that interrupted their habitats; (iii) The sedimentary features of the two beds are distinctly different, which suggests that different processes led to their accumulation. The thick allobiostrome exhibits a clast-supported texture across its whole

profile, and shows no vertical variation in terms of stromatoporoid size, shape, or preservation. Despite its smaller thickness, the parabiostrome is internally more complex, and exhibits vertical size, shape, and preservation sorting of the stromatoporoid skeletons. The allobiostrome represents a single act of deposition, whereas the parabiostrome resulted from a more complicated multiphase process. The absence of sorting, clast supported textures, and good preservation state of the skeletons agree with a tsunami interpretation of the allobiostrome. Intermittent storm activity is sufficient to explain the sedimentary features of the parabiostrome, including the sorting patterns of stromatoporoid skeletons within the bed.

## Author Contributions

**Conceptualization:** Piotr Łuczyński.

**Data curation:** Piotr Łuczyński.

**Formal analysis:** Piotr Łuczyński.

**Methodology:** Piotr Łuczyński.

**Project administration:** Piotr Łuczyński.

**Supervision:** Piotr Łuczyński.

**Writing – original draft:** Piotr Łuczyński.

**Writing – review & editing:** Piotr Łuczyński.

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
