## [Decision Letter · Decision Letter 0]

18 Feb 2022

PONE-D-21-38311Tsunamites versus tempestites – various types of redeposited stromatoporoid beds in the Devonian of the Holy Cross Mountains (Poland): a case study from the Ołowianka QuarryPLOS ONE

Dear Dr. Łuczyński,

Thank you for submitting your manuscript to PLOS ONE. After careful consideration, we feel that it has merit but does not fully meet PLOS ONE's publication criteria as it currently stands. Therefore, we invite you to submit a revised version of the manuscript that addresses the points raised during the review process.

Please follow the reviewers' suggestions, to make some corresponding revisions to the text and figures.

We look forward to receiving your revised manuscript.

Kind regards,

Jinzhuang Xue

Academic Editor

PLOS ONE

Journal Requirements:

2.  We note that this study addresses research carried out in an abandoned quarry. In your Methods section, please provide additional information regarding the permits you obtained from the landowner for carrying out the work on this land. Please ensure you have included the full name of the authority that approved the field site access and, if no permits were required, a brief statement explaining why.

(The studies were financed by the Faculty of Geology, University of Warsaw.)

(The study was financed from the funds of the Faculty of Geology, University of Warsaw

he funders had no role in study design, data collection and analysis, decision to publish, or preparation of the 

5. We noted in your submission details that a portion of your manuscript may have been presented or published elsewhere.

( The manuscript has been presented to dr Mikołaj Zapalski, PLOS One editor, who’s preliminary opinion was that it is suitable for submission in your Journal.) 

Please clarify whether this publication was peer-reviewed and formally published. If this work was previously peer-reviewed and published, in the cover letter please provide the reason that this work does not constitute dual publication and should be included in the current manuscript.

5. We note that Figures 1, 2, and 3 in your submission contain map images which may be copyrighted. All PLOS content is published under the Creative Commons Attribution License (CC BY 4.0), which means that the manuscript, images, and Supporting Information files will be freely available online, and any third party is permitted to access, download, copy, distribute, and use these materials in any way, even commercially, with proper attribution. For these reasons, we cannot publish previously copyrighted maps or satellite images created using proprietary data, such as Google software (Google Maps, Street View, and Earth). For more information, see our copyright guidelines: http://journals.plos.org/plosone/s/licenses-and-copyright.

a. You may seek permission from the original copyright holder of Figures 1, 2, and 3 to publish the content specifically under the CC BY 4.0 license.  

Reviewers' comments:

Reviewer's Responses to Questions

**Comments to the Author**

1. Is the manuscript technically sound, and do the data support the conclusions?

Reviewer #1: Yes

Reviewer #2: Yes

2. Has the statistical analysis been performed appropriately and rigorously? 

Reviewer #1: Yes

Reviewer #2: Yes

3. Have the authors made all data underlying the findings in their manuscript fully available?

Reviewer #1: Yes

Reviewer #2: Yes

4. Is the manuscript presented in an intelligible fashion and written in standard English?

Reviewer #1: Yes

Reviewer #2: No

5. Review Comments to the Author

Reviewer #1: This manuscript is a very detailed taphonomy work on stromatoporoids. The author describes the allobiostrome and parabiostrome of Kowala Formation and attributes the causes to be tsunamites and tempestites, respectively. The description is very specific way, with sufficient data, and discussion is in a logical way. The manuscript can be published by polishing the language and correct some grammer mistakes, such as "thought (line 57)". I feel that some paragraphs, especially in the morphometrical analysis and results, can be reduced or combined, so that the entire manuscript is more clear in structure. In addition, the thin section photos are not quite satisfactory and can be improved by adjusting contrasts and brightness. The field photos can be replaced with colors. Some figures, such as figure 1 and 2, and figure 3 and 6 can be combined or adjusted.

Reviewer #2: This work is interesting, and is significant for the study of tempestites. However, there are some questions in the manuscript.

Some English expressions are too complicated, such as Line 647-648, so concise language is required.

The demonstration on the relationship between shape and growth form of stromatoporoid is unclear.

The conclusion is disorganization.

Line 83-85. This is especially intriguing, taking onto account that flat pebble conglomerates of an alleged tsunami genesis accompany also the stromatoporoid parabiostromes of Podolia. “Taking into account” may be more appropriated.

Line 341-342. Relative explains of Figure 7 G, H are missing. The sentence of “The wall on A is approximately 4m high, the visible part of the wall on D is approximately 3m high” is very confused.

Figure 7. The explanations of scale bars are missing.

Line 383-385. There are no G and H in Figure 8.

Line 393-396. Results of morphometrical analysis of the skeletons, combined with taphonomical and sedimentological data, were interpreted in terms of the stromatoporoid growth environments, and of the sedimentary conditions than lead to their accumulations. “Than” may be incorrect.

Line 428-431. This indicates than in the majority of cases the high shape profile describes not only the whole skeleton, but also the growth form (living surface profile), representing the part of the skeleton protruding from the sediment above the sea bottom at the final stage of the stromatoporoids growth. “Than” may be incorrect.

Line 541. “autobiostrome” may be incorrect.

Line 823-825. T ‒ attributes typical and/or expected f 823 or tsunamites, T ‒ attributes unexpected for tsunamites, S – attributes typical and/or expected for storm deposits (tempestites), S ‒ attributes unexpected for storm deposits. This sentence is repeated.

The explanation of “----------” in Table 2 is missing.

Individual punctuations are wrong, such as Line 887.

6. PLOS authors have the option to publish the peer review history of their article (what does this mean?). If published, this will include your full peer review and any attached files.

Reviewer #1: **Yes: **Kun Liang

Reviewer #2: No

---

## [Author Response · Author response to Decision Letter 0]

17 Mar 2022

I. Points raised by the academic editor

The manuscript has been checked, and all style requirements, including file naming, are being met. 

2. In your Methods section, please provide additional information regarding the permits you obtained from the landowner for carrying out the work on this land ….. if no permits were required, a brief statement explaining why.

The following statement was added in the “Materials and methods” section: Fieldwork was conducted in the abandoned Ołowianka Quarry. Exploitation in the quarry ceased in the 1980s: today, it is state owned, barren land with free access to the general public. According to Polish law, no permits are required to perform geological studies in such an area

Corrected. 

4. Please remove any funding-related text from the manuscript and let us know how you would like to update your Funding Statement.

The funding-related text has been removed from the manuscript. The current funding statement is correct and does not need to be updated 

5. We noted in your submission details that a portion of your manuscript may have been presented or published elsewhere. ( The manuscript has been presented to dr Mikołaj Zapalski, PLOS One editor, who’s preliminary opinion was that it is suitable for submission in your Journal.) Please clarify whether this publication was peer-reviewed and formally published. If this work was previously peer-reviewed and published, in the cover letter please provide the reason that this work does not constitute dual publication and should be included in the current manuscript.

The work has not been peer reviewed or published in any way. The mentioning of presenting the paper to dr Mikołaj Zapalski, PLOS One editor, meant only that I asked his personal advice, based on his experience as a PLOS editor, if the topic and volume of the paper generally fits the expectations and requirements of your journal. He has just glanced the text, did not check it, or make any corrections. 

The only data I have collected, which eventually can be examined by other workers, other than field photographs presented in the figures, are the thin plates illustrated on figure 7. All other discussed data are observations made in field.

Therefore, please put the following information in the Data Availability Statement “The studied thin plates illustrated on figure 7 are deposited and available in the S.J. Thugutt Geological Museum, Faculty of Geology, University of Warsaw, Poland (MWGUW) under the inventory number: MWGUW 009779. 

7. We note that Figures 1, 2, and 3 in your submission contain map images, which may be copyrighted. Please upload the completed Content Permission Form or other proof of granted permissions as an "Other" file with your submission. In the figure caption of the copyrighted figure, please include the following text: “Reprinted from [ref] under a CC BY license, with permission from [name of publisher], original copyright [original copyright year].”

Copyright permissions have been obtained from the copyright holders (files “Other”) . However, figures 1, 2, and 3 have been changed, according to the suggestions of one of the reviewers (figures 1 and 2 have been merged into one). None of these figures simply copy the figures from previous papers (including maps), but are constructed by myself based on previous papers, with considerable changes. Therefore, in the captions I did not write “reprinted from…..” but “modified after…..”

8. Please review your reference list to ensure that it is complete and correct. 

The reference list has been thoroughly checked 

II. Points raised by reviewer #1

1. Polishing the language and correct some grammar mistakes

The whole text has been corrected by a native speaker 

2. Some paragraphs, especially in the morphometrical analysis and results, can be reduced or combined, so that the entire manuscript is more clear in structure. 

The structure has been left as previously

3. Thin section photos are not quite satisfactory and can be improved by adjusting contrasts and brightness. 

Thin section photos have been improved (figure 6, according to new numeration) 

4. The field photos can be replaced with colors.

Done 

 5. Some figures, such as figure 1 and 2, and figure 3 and 6 can be combined or adjusted.

Figures 1 and 2 have been merged into one figure. Figures 3 and 6 ( 2 and 5 according to new numeration) remain separate

III. Points raised by reviewer #2

1. Some English expressions are too complicated …. concise language is required. 

The whole text has been corrected by a native speaker. All the specific points, with reference to particular lines (647-648, 83-85, 393-396, 428-431, 541, 823-825, and 887) have been corrected. 

2. The demonstration on the relationship between shape and growth form of stromatoporoid is unclear. 

I hope now it is clear. The topic has been discussed in a number of papers cited in the text 

3. The conclusion is disorganization.

I don’t understand this comment 

4. Line 341-342. Relative explains of Figure 7 G, H are missing. The sentence of “The wall on A is approximately 4m high, the visible part of the wall on D is approximately 3m high” is very confused.

 Figures 7 and 8 have been switched. Now the captions comply with the figures 

5. Figure 7. The explanations of scale bars are missing. 

Figures 7 and 8 have been switched. Scale bars on new figure 6 have been explained 

6. The explanation of “----------” in Table 2 is missing. 

This simply means that this particular feature does not apply here. Some features can be interpreted in terms of stromatoporoid growth environment, some in terms of transport and final accumulation, but not all in terms of both these aspects. I don’t feel this needs explanation in the table captions.

---

## [Editor Report · Decision Letter 1]

28 Apr 2022

Tsunamites versus tempestites: various types of redeposited stromatoporoid beds in the Devonian of the Holy Cross Mountains (Poland), a case study from the Ołowianka Quarry

PONE-D-21-38311R1

Dear Dr. Łuczyński,

We're pleased to inform you that your manuscript has been judged scientifically suitable for publication and will be formally accepted for publication once it meets all outstanding technical requirements.

Within one week, you'll receive an e-mail detailing the required amendments. When these have been addressed, you'll receive a formal acceptance letter and your manuscript will be scheduled for publication.

If your institution or institutions have a press office, please notify them about your upcoming paper to help maximize its impact. If they'll be preparing press materials, please inform our press team as soon as possible -- no later than 48 hours after receiving the formal acceptance. Your manuscript will remain under strict press embargo until 2 pm Eastern Time on the date of publication. For more information, please contact onepress@plos.org.

Kind regards,

Jinzhuang Xue

Academic Editor

PLOS ONE

Additional Editor Comments (optional):

I am pleased to inform you that your manuscript has been accepted for publication. You have made thorough revisions to make the paper more clear and concise.  Congratulations!
---

## [Editor Report · Acceptance letter]

4 May 2022

PONE-D-21-38311R1 

Tsunamites versus tempestites: various types of redeposited stromatoporoid beds in the Devonian of the Holy Cross Mountains (Poland), a case study from the Ołowianka Quarry 

Dear Dr. Łuczyński:

I'm pleased to inform you that your manuscript has been deemed suitable for publication in PLOS ONE. Congratulations! Your manuscript is now with our production department. 

Kind regards, 

on behalf of

Dr. Jinzhuang Xue 

Academic Editor

PLOS ONE